# Face mask fit hacks: Improving the fit of KN95 masks and surgical masks with fit alteration techniques

**Eugenia O'Kelly**[1]*, **Anmol Arora**[2], **Sophia Pirog**[3], **Charlotte Pearson**[4], **James Ward**[1], **P. John Clarkson**[1]

1 Department of Engineering, University of Cambridge, Cambridge, United Kingdom, 2 School of Clinical Medicine, University of Cambridge, Cambridge, United Kingdom, 3 Feinberg School of Medicine, Northwestern University, Evanston, IL, United States of America, 4 Lundquist College of Business, University of Oregon, Eugene, OR, United States of America

* eo339@cam.ac.uk

**Data Availability Statement:** All relevant data are within the manuscript.

**Funding:** The authors received no specific funding for this work.

## Abstract

### Introduction

During the course of the COVID-19 pandemic, there have been suggestions that various techniques could be employed to improve the fit and, therefore, the effectiveness of face masks. It is well recognized that improving fit tends to improve mask effectiveness, but whether these fit modifiers are reliable remains unexplored. In this study, we assess a range of common "fit hacks" to determine their ability to improve mask performance.

### Methods

Between July and September 2020, qualitative fit testing was performed in an indoor living space. We used quantitative fit testing to assess the fit of both surgical masks and KN95 masks, with and without 'fit hacks', on four participants. Seven fit hacks were evaluated to assess impact on fit. Additionally, one participant applied each fit hack multiple times to assess how reliable hacks were when reapplied. A convenience of four participants took part in the study, three females and one male with a head circumference range of 54 to 60 centimetres.

### Results and discussion

The use of pantyhose, tape, and rubber bands were effective for most participants. A pantyhose overlayer was observed to be the most effective hack. High degrees of variation were noted between participants. However, little variation was noted within participants, with hacks generally showing similar benefit each time they were applied on a single participant. An inspection of the fit hacks once applied showed that individual facial features may have a significant impact on fit, especially the nose bridge.

**Competing interests:** The authors have declared that no competing interests exist.

## Conclusions

Fit hacks can be used to effectively improve the fit of surgical and KN95 masks, enhancing the protection provided to the wearer. However, many of the most effective hacks are very uncomfortable and unlikely to be tolerated for extended periods of time. The development of effective fit-improvement solutions remains a critical issue in need of further development.

## Introduction

During the COVID-19 pandemic, shortages of personal protective equipment (PPE) have resulted in members of the public having no choice other than to wear poorly fitting face masks, including surgical and KN95 masks, which are the focus of this study. Proper fit has been noted as a primary factor in determining the effectiveness of face masks, but the masks available to the public often suffer from poor fit [1]. In an effort to improve the protection such masks offer, several alterations, a.k.a. fit hacks, have emerged in an attempt to improve fit. While these fit hacks have garnered widespread attention and media coverage, the impact of most hacks on fit remain largely untested. Such fit hacks include the use of pantyhose over the mask and altering the mask shape by knotting the ear bands [2, 3]. Early research suggests that these fit hacks are effective at improving the fit of masks with another research group testing several techniques; however, in their experiments only one individual was tested [4].

There are two key factors that determine mask effectiveness: the filtration efficiency of the mask material and the fit of the mask. Better fitting masks offer fewer gaps between the wearer's face and the edges of the mask, ensuring that inhaled air is actually filtered. Air tends to take the path of least resistance; therefore, if there are small gaps around a high resistance mask, air will tend to travel through them, thus bypassing filtration of the mask material [5]. Reducing these gaps, and thus improving fit, enhances not only the protection the mask can offer the wearer from airborne particles, but it also offers greater protection to the public. In addition, better fit for the wearer has been theorized to reduce the size of SARS-CoV-2 inoculum, thus potentially eliminating or reducing the severity of an infection [6].

Unfortunately, little research has been done to assess the effectiveness of these techniques. Early research, seeking to validate a protocol for measuring mask performance, has suggested that a nylon over-layer worn over a face covering may enhance the performance of masks. However, the same data also suggests that improvement techniques may heterogeneously affect different types of face coverings [2]. This remains an understudied area of research with a notable shortage of multi-participant studies. This study aims to answer the research question: "Do simple fit hacks actually improve the fit of face masks?" We address this by exploring the quantitative fit score of several masks with and without a range of fit hacks applied. We evaluate both (1) whether fit hacks are effective, and (2) whether their effectiveness varies between individuals. In doing so, this research seeks to discover which, if any, techniques may be used to improve the fit of face masks aimed primarily for use by the general public.

## Materials and methods

### Quantitative fit testing method

There are two established methods used to assess the fit of face masks: quantitative fit testing and qualitative fit testing [7, 8]. Quantitative fit testing is the most robust and accurate method, providing a nuanced measurement of the degree of fit. Qualitative fit testing, on the other

hand, involves spraying a compound in the ambient air and testing whether or not the wearer of the mask can taste it whilst wearing a mask. This is a less reliable method of assessing fit as it often relies on subjective judgements of taste and cannot quantify the fit. As such, quantitative fit testing was utilized in this study to evaluate the efficacy of fit hacks.

During quantitative fit testing, air samples are taken to continuously measure the concentrations of particles both inside and outside of a donned mask. The concentration of particles outside the mask are compared via an industry standard formula, to generate a fit factor score [9, 10]. The formula is:

$$\text{Fit Factor} = (\text{CB} + \text{CA})/2\text{CR}$$

CB = concentration B, concentration outside the mask before the respiratory sample

CA = concentration A, concentration outside the mask after the respiratory sample

CR = concentration R, concentration inside the mask

This fit factor is a numerical score of how well the mask fits the wearer; meaning the fewer particles that make their way into the mask, the better the fit. As fit factor is particle concentration outside divided by particle concentration inside, a mask with a fit factor of 100 provides air 100 times cleaner than outside air and a mask with a fit factor of 10 provides air 10 times leaner than the outside air [11]. Higher scores indicate better fit while low scores indicate a poor fit. N95 or FFP3 masks must score at least 100 to be considered to have adequate fit by OSHA standards [10].

Quantitative fit tests were conducted using a Portacount, TSI, Minnesota, model 8038 + capable of evaluating masks with less than 99% efficiency [10]. Manufacturer specifications indicate readings are accurate in most cases within +/- 10% of fit factor. This is represented in error bars unless otherwise stated. Particles ranging from 0.02 micrometers to over 1 micrometer were measured. A particle generator from TSI, Minnesota, Model 8026 was used to atomize sodium chloride (NaCl) particles during testing. KN95 respirator was a Zhong Jian Le KN95 respirator, manufactured by Chengde Technology Co LTD, China and certified according to Chinese standard GB2626-2006.

Four participants took part in the study, three female and one male. Ethical approval for this study was given by the Cambridge University Department of Engineering Ethics Committee. Participants completed seven activities intended to reproduce a range of occupational activity according to the original OSHA protocol 29CFR1910.134 Appendix A, which has since been modified to be shorter [12, 13]. We chose to use the original protocol, which places emphasis on activities common in the general public, such as heavy and light breathing and tests the mask's ability to maintain fit after exaggerated facial expressions [12]. All activities in the new 29CFR1910.134 Appendix A-2 table are included in the old test protocol [13]. Each test subject performed the following activities: normal breathing, heavy breathing, turning head side to side, nodding head up and down, talking, bending over, smiling/grimacing (score excluded), and normal breathing after mask fit was stressed by smiling/grimacing. Each test collected 7 minutes 15 seconds of mask-wearing data [12]. Due to the length of the study, over two hours per participant, participants were allowed to sit during certain sections of the test.

## Fit hacks

A range of fit hacks, taken from the internet and from observing the public, were tested with two masks: a KN95 mask and a surgical mask. Fit scores of the mask without a fit hack were

compared with the fit scores once the fit hack was applied in order to determine the fit hack's impact.

In this study, we focus on KN95 and surgical masks as both masks are widely available and, if fit properly, can offer a high degree of filtration. KN95 masks promise similar filtration benefits to N95 masks; however, unlike N95 masks, the public's access to KN95 masks has been less restricted by this COVID-19 pandemic. While KN95 masks offer significant filtration potential, their poor fit on many individuals negates the potential benefit of these masks [14]. Surgical masks are in common use and, if constructed out of the proper materials, can offer a very high filtration ability. The surgical masks in this study were secured by earloops behind the ear and are also referred to as procedural masks.

Seven fit hacks were tested, as shown in Fig 1 and as described below:

- Tape: The edges of the mask are sealed with cloth tape. Care was taken to mould the tape to the face and to seal any gaps between the skin and the edge of the mask.

- Stuffed Gaps: First aid gauze was used to fill visible gaps in the mask, until no visible space between skin and mask remained.

- Mummy: A roll of first-aid gauze was used to tightly bind the mask to the face, pressing the edges of the mask to the skin of the face.

- Pantyhose: Two brands of pantyhose were separately placed over the head to press the mask into place, a method first proposed and tested by Mueller et al. [2].

- Knotting Ear Loops: To make a large mask fit a smaller face, an overhand knot was made of the ear loop elastic near the mask. This hack gained media attention when dentist Dr. Olivia Cui posted a video of herself performing the hack on TikTok [3].

- Rubber Bands: In a hack proposed by Apple engineers, three rubber bands are used to create a 'brace' [15].

Four participants were tested to determine if the benefits incurred by the application of a fit hack differed significantly between individuals. Participants are identified here by their respective gender and age. Participant F-29 had the smallest head size with a circumference of 54cm. Participants F-51 and F-18 had a circumference of 55cm and 56cm respectively. M-20 had the largest head size, with a circumference of 60cm.

All hacks were tested with two exceptions. These two exceptions were a result of an inability of the wearer to apply the fit hack. In one case, a participant was unable to fit the KN95 mask with ear bands tied, as the adjustment caused the wire of the mask to rub against a painful sore. Another participant was unable to fit pantyhose Brand A over his head as it was too tight and unable to accommodate the participant's head size. In these two cases, the hack was not tested.

To assess the reliability of these hacks, a further experiment was conducted on one participant. This participant applied each hack five separate times to determine whether changes in fit as a result of the hack were reproducible when the mask is doffed and donned.

## Setting

Experiments were performed indoors in a clean environment. Temperatures and relative humidity were not precisely measured at the time of experimentation, but are estimated to be between 65 and 75 oF and 65% and 70% respectively. As fit factor is calculated by comparing the relative concentrations of particles inside and outside of the mask, we do not expect this to

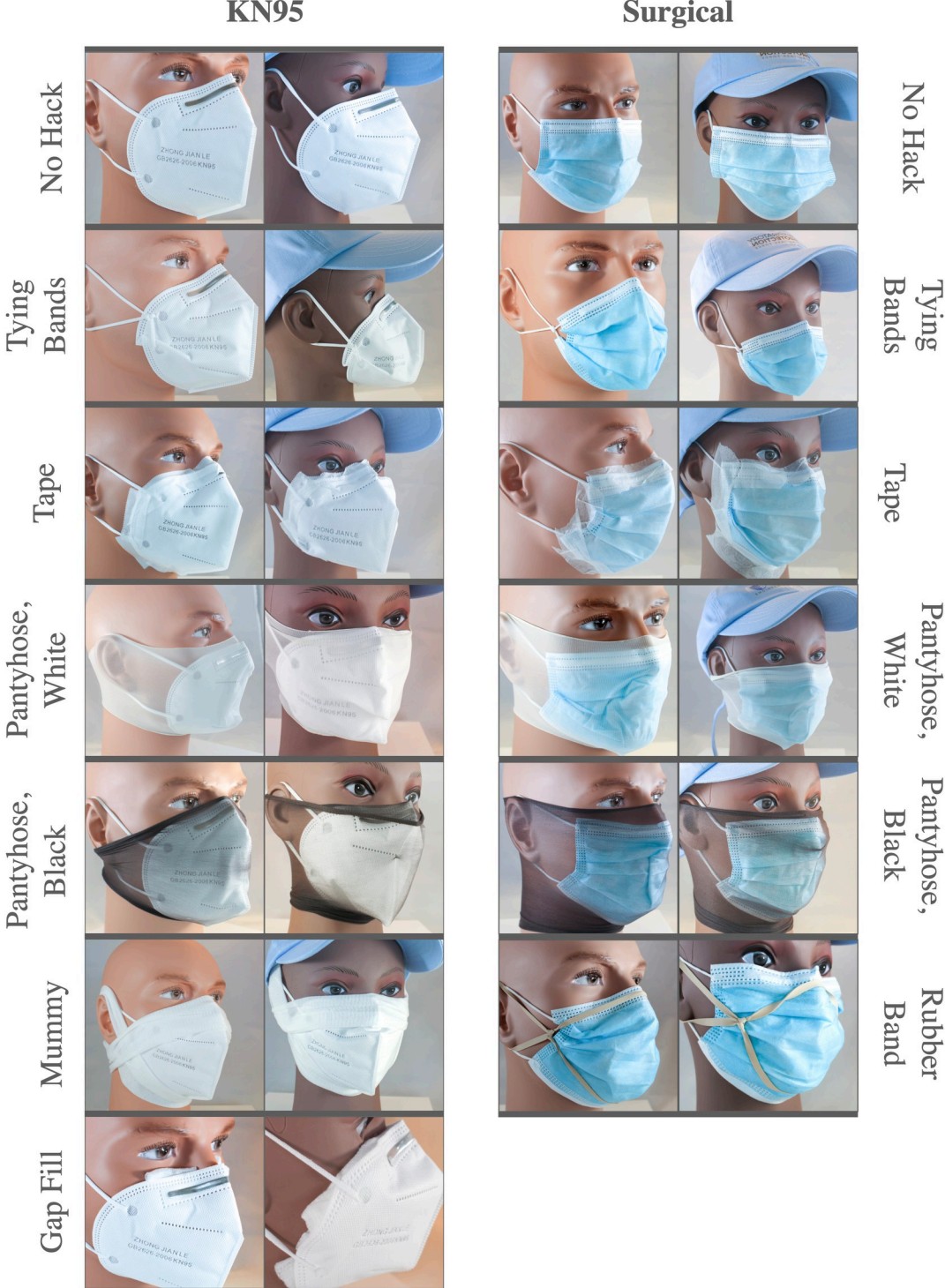

**Fig 1. Pictures of the tested fit hacks applied to KN95 and surgical masks on two mannequin heads for display.** The male mannequin head is the size of an average male, while the female represents the size of a small female head or young teen head.

influence the results. However, it should be noted that particle counts may vary with temperature and humidity.

## Results

### KN95 masks

As shown in Fig 2, the pantyhose and tape hacks were most effective at improving fit in KN95 masks, although significant variation between participants occurred. The use of pantyhose produced improvement in the fit of KN95 masks depending on brand, with an average fit factor improvement of 27.7 for Brand A but only 5 for Brand B. Using tape to seal the edges of the KN95 mask improved fit factor significantly, with an average improvement of 14.7. The use of gauze to seal gaps offered a minor improvement of 2.8 while using gauze to bind the mask to the face via the mummy hack improved fit by only 1.6. Tying ear bands resulted in an average improvement of only 0.8.

### Surgical masks

Pantyhose and cloth tape around the edges of the mask improved fit significantly when applied to surgical masks (see Fig 3). Pantyhose proved effective in improving the fit factor of surgical masks, with an average improvement of 7.2 for Brand A and 4.9 for Brand B. Tape provided a similar average improvement of 4.8. The least effective hacks were the use of rubber bands, with an average improvement of 2.5, and tying ear bands, with an average improvement of 2.5.

### Reproducibility of fit improvements

The results of a single hack applied multiple times to the same individual were highly consistent in most cases (see Fig 4). The exception to this was in the application of the pantyhose and cloth tape hacks to KN95 masks. In the case of the tape, this high degree of variation was likely due to minor variations in the degree to which the tape was able to seal the edges of the mask. As previous studies have shown, even small gaps greatly affect the performance of a mask [14]. Similarly, the high degree of variation between applications of the pantyhose hack may indicate that the hack has the potential to tightly seal the mask to the face, but that this potential is not always achieved during every application. Applying the pantyhose over the mask was difficult as it was tight-fitting. Sometimes its application would disturb the placement of the mask

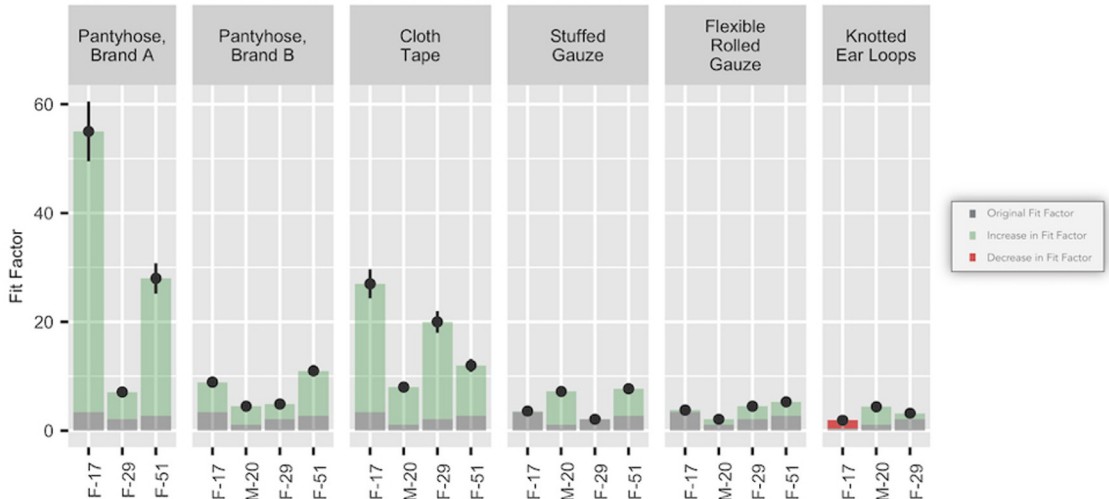

**Fig 2. Fit hacks applied to KN95 masks.** Gray portions of the bars indicate the performance without the application of a fit hack. Green portions of the bars indicate the amount of improvement. The red portion of the bar indicates the amount of decrease in performance.

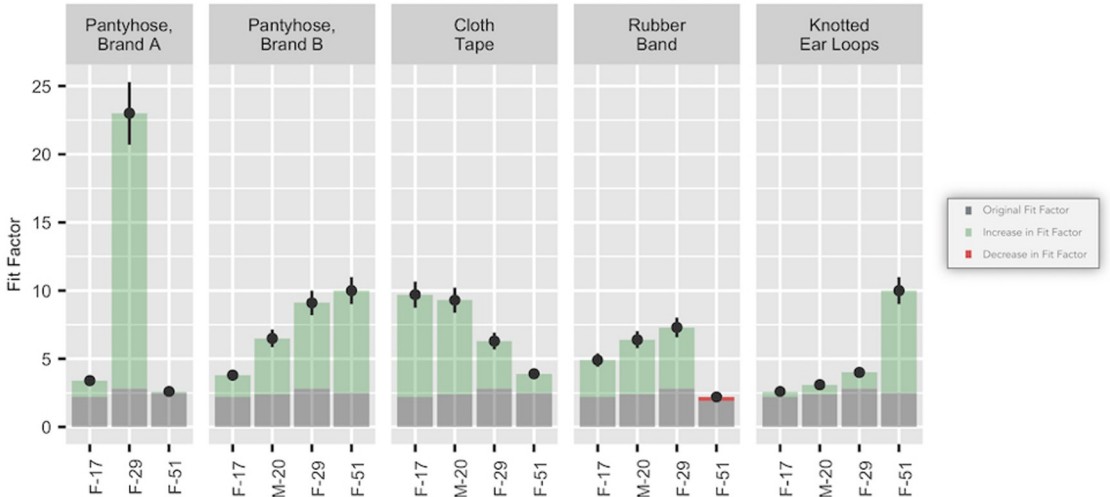

**Fig 3. Fit hacks applied to surgical masks.** Gray portions of the bars indicate the performance without the application of a fit hack. Green portions of the bars indicate improvement gained with the application of a fit hack. The red portion of the bar indicates a decrease in performance from applying the fit hack.

such that it became tilted or off-centre. Changes in hairstyle also affected the tightness of the pantyhose over the mask, which is a key determinant of fit. Similarly, fit depends on the size and type of pantyhose used. Furthermore, fit factor is not scaled linearly with filtration efficiency, and, within higher fit factor values, small changes in filtration efficiency can be represented as larger changes in fit factor.

## Discussion

With few exceptions, fit hacks improved the fit factor of both surgical and KN95 masks for all participants. The pantyhose and tape hacks proved to be the most effective hack for both KN95 and surgical masks, with the rubber band hack also showing some promise for surgical masks.

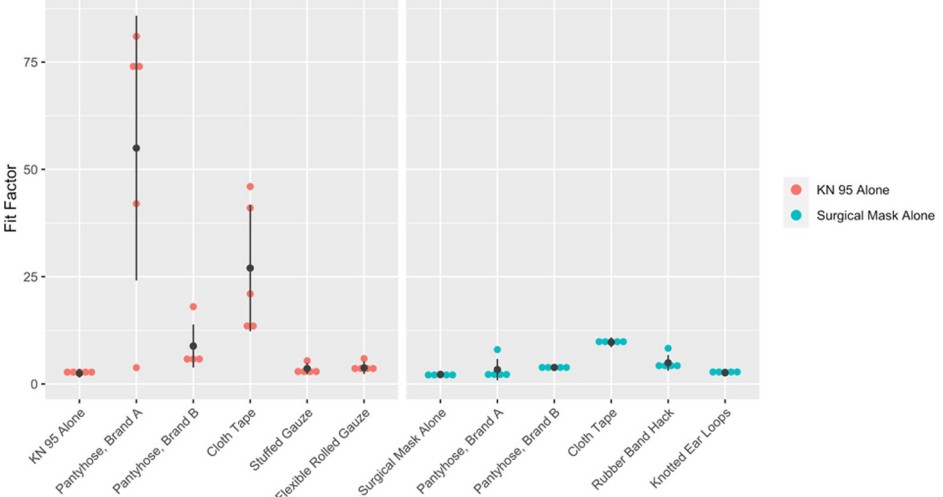

**Fig 4. A single fit hack applied five times to one individual demonstrates how reliable the application of a hack is at improving fit factor.**

The benefits of each hack differed greatly according to the participant, sometimes in surprising ways. For example, it was expected that fit hacks to make a mask smaller would not significantly benefit participants with large heads. While this proved true for surgical masks, with the individuals with the two smallest heads benefiting the most, it did not hold true for KN95 masks.

An inspection of the fit hacks once applied showed that individual facial features may have a significant impact on fit. For example, a visual inspection of the hacks when applied to participants showed that the nose bridge prevented some hacks from contouring to the sides of the nose, a more significant issue for those with prominent noses (see Fig 5). This type of issue may help explain the lower reliability of the pantyhose hack, because the placement of the pantyhose over the nose bridge can create a gap between infraorbital skin and the overlying mask.

Discomfort was an issue with many of the hacks. The most discomfort was reported with the pantyhose and the rubber band hacks. The rubber band hack was found to put painful pressure on the ears and face, going so far as to hinder circulation to the ears for some participants. The pantyhose caused high levels of discomfort as well as issues speaking and occasional obstruction of the eyes. The use of tape was reported to be comfortable while worn, but moderate to high levels of discomfort accompanied the tape's removal. These observations indicate that although a tighter fit provides greater protection, this may be at the expense of the wearer's comfort.

The most effective fit hack was the use of pantyhose. Placing pantyhose over the mask and head proved an effective way to improve fit, though it should be noted that the brand of pantyhose used was found to have a significant impact on the benefits incurred. We used large sizes of Brand A and Brand B for our experiment. A thigh section was cut to be placed around the head. The material of Brand A was found to be tighter and less flexible. Overall it created an improvement which was, on average, approximately two times greater than that of Brand B. The use of pantyhose generated an improvement for all testers, although one tester was unable to fit the pantyhose over their head. The reliability of the pantyhose when applied multiple times on one tester was variable.

The use of tape to seal the edges of a mask was the second most effective hack. Participants who benefited from the hack the most had an assistant to ensure the tape was correctly placed and no gaps were present. How cloth tape would seal the mask over longer periods of time is

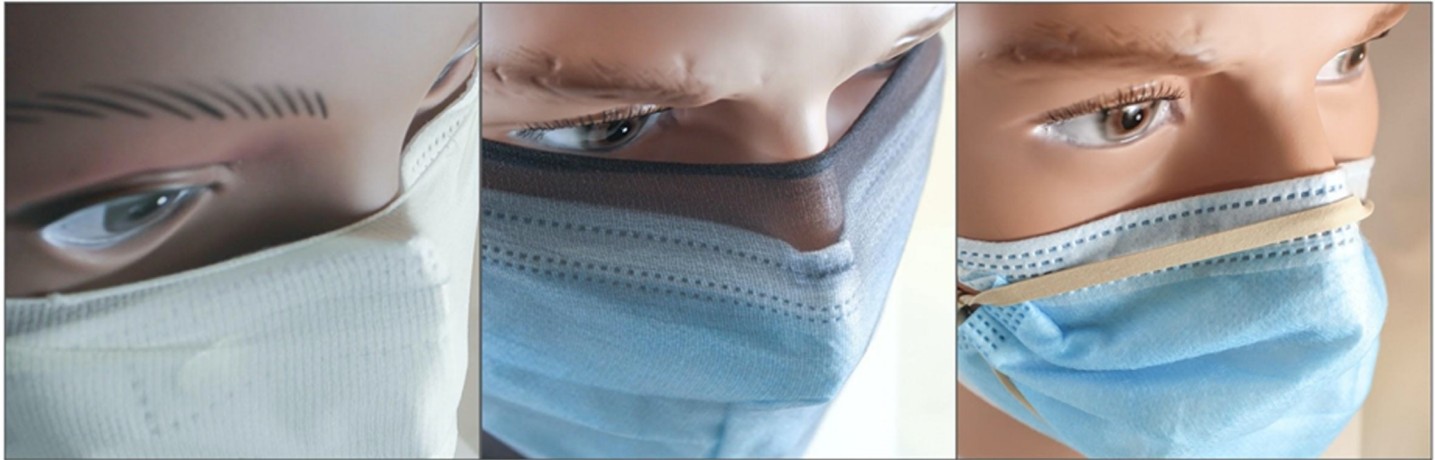

**Fig 5. Many fit hacks were unable to fix fit issues around the nose bridge.** This impacted wearers with more prominent noses and provided an example of one facial feature critical to the efficacy of fit hacks.

unknown as sweat or movement would be expected to degrade the seal of all but the most flexible tape. As expected, the reliability of this hack varied, likely due to how accurately the tape was applied and if the tape became loose in an area at any point during testing.

Higher fit scores were achieved when hacks were applied to the KN95 mask. Although KN95 masks have a high filtration efficiency, the fit is often poor and our results indicate that using KN95 masks with fit hacks can potentially provide high levels of protection. One participant was unable to fit the KN95 mask with ear bands tied. Another participant was unable to fit one brand of pantyhose over his head without the pantyhose tearing. In these two cases, the problematic hack was not tested.

Our findings are consistent with those of Clapp et al (2020), who tested five similar fit hacks, including tying ear loops and applying nylon hosiery over the mask [4]. The nylon overlayer was their most effective fit hack. Although, they noted that whilst the techniques did improve mask fit, they were not always comfortable or practical for the wearer.

## Conclusions

Maximizing the protection a mask provides oneself and others rests heavily on improving the fit of the masks. Our results indicate that there is potential for fit hacks to improve the fit of masks, by sealing the edges of the mask or pressing the masks tightly to the face. We would recommend that new mask designs should focus on ensuring that the edges of the mask are firmly in contact with the face.

However, whilst the study does indicate that hacks may be successful, much work remains to be done to create comfortable, effective fit improvements. Many of the most effective fit hacks were so uncomfortable as to be unusable in some cases. Overall, whilst we found that fit hacks did generally improve mask performance, it is difficult to predict the effects for a given individual or face type.

We hope these results will be of benefit to designers in order that they may improve masks and mask fitting devices, as well as members of the public seeking to improve their own masks. The hacks tested are all accessible to the general public. As surgical masks and KN95 masks are commonly worn by healthcare professionals, these findings may assist them improve the protection they are obtaining from their masks. Further research efforts should seek to validate these findings, test a wider variety of fit hacks and expand the range of masks tested to include fabric face coverings.

## Author Contributions

**Conceptualization:** Eugenia O'Kelly, James Ward, P. John Clarkson.

**Data curation:** Eugenia O'Kelly, Sophia Pirog, Charlotte Pearson.

**Formal analysis:** Eugenia O'Kelly, Sophia Pirog.

**Investigation:** Eugenia O'Kelly, Anmol Arora, Charlotte Pearson.

**Methodology:** Eugenia O'Kelly.

**Project administration:** Eugenia O'Kelly.

**Resources:** Eugenia O'Kelly.

**Supervision:** Eugenia O'Kelly, James Ward, P. John Clarkson.

**Validation:** Eugenia O'Kelly.

**Visualization:** Eugenia O'Kelly, Sophia Pirog.

**Writing – original draft:** Eugenia O'Kelly, Anmol Arora.

**Writing – review & editing:** Eugenia O'Kelly, Anmol Arora.

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
