## [Decision Letter · Decision Letter 0]

21 May 2021

PONE-D-21-11954

Face Mask Fit Hacks: Improving the Fit of KN95 Masks and Surgical Masks with Fit Alteration Techniques

PLOS ONE

Dear Dr. O'Kelly,

Thank you for submitting your manuscript to PLOS ONE. After careful consideration, we feel that it has merit but does not fully meet PLOS ONE’s publication criteria as it currently stands. Therefore, we invite you to submit a revised version of the manuscript that addresses the points raised during the review process.

I received comments and recommendations from two reviewers.Please well address these comments and improve the quality of this manuscript.

We look forward to receiving your revised manuscript.

Kind regards,

Jianguo Wang, PhD

Academic Editor

PLOS ONE

Journal Requirements:

2. Please provide additional details regarding participant consent. In the ethics statement in the Methods and online submission information, please ensure that you have specified (i) whether consent was informed and (ii) what type you obtained (for instance, written or verbal, and if verbal, how it was documented and witnessed). If your study included minors, state whether you obtained consent from parents or guardians. If the need for consent was waived by the ethics committee, please include this information.

Reviewers' comments:

Reviewer's Responses to Questions

**Comments to the Author**

1. Is the manuscript technically sound, and do the data support the conclusions?

Reviewer #1: Yes

Reviewer #2: Yes

2. Has the statistical analysis been performed appropriately and rigorously? 

Reviewer #1: Yes

Reviewer #2: Yes

3. Have the authors made all data underlying the findings in their manuscript fully available?

Reviewer #1: Yes

Reviewer #2: Yes

4. Is the manuscript presented in an intelligible fashion and written in standard English?

Reviewer #1: Yes

Reviewer #2: Yes

5. Review Comments to the Author

Reviewer #1: During the course of the COVID-19 pandemic, there have been suggestions that various techniques could be employed to improve the fit and, therefore, the effectiveness of face masks. It is well recognized that improving fit tends to improve mask effectiveness, but whether these fit modifiers are reliable remains unexplored. This study used quantitative fit testing to assess the fit of both surgical masks and KN95 masks, with and without ‘fit hacks’, on four participants. Seven fit hacks were evaluated to assess impact on fit. Additionally, one participant applied each fit hack multiple times to assess how reliable hacks were when reapplied. The research of this paper is very important and useful. It can be used to effectively improve the fit of surgical and KN95 masks. The detailed comments are presented below:

1. In Fit hacks: All hacks were tested with two exceptions. One participant was unable to fit the KN95 mask with ear bands tied, as the adjustment caused the wire of the mask to rub against a sore. Another participant was unable to fit pantyhose Brand A over his head. In these two cases, the hack was not tested. Can you provide more details?

2. To assess the reliability of these hacks, a further experiment was conducted on one participant. This participant applied each hack five separate times to determine whether the changes in fit from the hack were reproducible when the mask is doffed and donned. The authors say five separate times to determine whether the changes. What's the reference?

3. The high degree of variation between applications of the pantyhose hack may indicate that the hack has the potential to tightly seal the mask to the face, but that this potential is not always achieved during every application. Furthermore, fit factor is not scaled linearly with filtration efficiency, and, within higher fit factor values, small changes in filtration efficiency can be represented as larger changes in fit factor. It is not sufficient to present only the results.

4. The conclusion is too simple, a good conclusion can be a restatement of the central idea, a summary of the research results or main points, some revelatory interpretation or consideration, and a prediction based on the research results? It is recommended to rewrite the conclusion.

Reviewer #2: The manuscript by O’Kelly and colleagues entitled “Face Mask Fit Hacks: Improving the Fit of KN95 Masks and Surgical Masks with Fit Alteration Techniques” aims to assess whether the common “fit hacks” improve the surgical and KN95 mask performance. The notion of improving facemask performance is still an important interventional strategy to reduce SARS-COV-2 transmission as parts of the world are still experiencing serious outbreaks and vaccination campaign still requires time to inoculate naïve populations. I believe this manuscript proves important information of improving mask efficiency and should be published. However, major revision is necessary to answer some questions.

1) Please note there is a difference between a surgical mask (tie) and a procedural mask (earloop). I believe the ones tested in this manuscript are procedural masks.

2) Please indicate the models and manufacturers of the masks tested.

3) The abstract could be more concise, especially the “Results and Discussion” section. I also recommend not using vague words, such as “most” and “many”, repeatedly. How many fit hacks are effective exactly?

4) Line 65. The claim is not accurate because there are other reasons driving the necessity to wear a “surgical” mask or a KN95 in the public. For example, one would prefer a “surgical” mask than N95 because of comfort? In addition, please explain why not testing cloth masks. They are also widely available to and used by the public.

5) Line 67. Besides fit, material type is also important in filtration efficiency.

6) The authors have talked a lot about the “fit” of mask and explained what the fit is at line 75. I would suggest expanding the explanation and mentioning it earlier so that it will help readers to understand.

7) Line 94-97. I think emphasizing the advantage of quantitative over qualitative fit testing here is not necessary. If the authors really think these are necessary, I believe they should belong to the discussion section.

8) Line 100. What exactly is the standard formula? I think providing the formula or equation will really help readers understand how the quantitative method works, including how the fit factor scores were assigned.

9) Line 106: does a Portacount generate real time data output, or just a summary of the whole test results? Is it only giving a “pass” or “fail” result?

10) Line 110: have the researchers monitored the relative humidity and temperature of the testing environment? I think reporting these data are important since the relative humidity seems to affect the particle count significantly.

11) Line 115: The test protocol involves breathing, talking, bending over, and turning the head. Which exact protocol in 29 CFR § 1910.134 was used? Table A-1, A-2, or A-3?

12) The data based on fit factor score is hard to understand without knowing how it was calculated and without a good comparing taget. For example, it will help if a N95 with good fit is tested (assuming >=95% particles were blocked from entry into the mask) and assigned a score.

13) Another common hack that could improve the mask performance is using a hook to tie the two earloops behind the head. Have you considered and tested this?

14) Line 274. I thought this paper was mainly to improve mask performance of the masks available and worn by the public. Healthcare workers seem to have no N95 shortage now, at least in the US.

6. PLOS authors have the option to publish the peer review history of their article (what does this mean?). If published, this will include your full peer review and any attached files.

Reviewer #1: No

Reviewer #2: No

---

## [Author Response · Author response to Decision Letter 0]

29 Jun 2021

Reviewer #1: During the course of the COVID-19 pandemic, there have been suggestions that various techniques could be employed to improve the fit and, therefore, the effectiveness of face masks. It is well recognized that improving fit tends to improve mask effectiveness, but whether these fit modifiers are reliable remains unexplored. This study used quantitative fit testing to assess the fit of both surgical masks and KN95 masks, with and without ‘fit hacks’, on four participants. Seven fit hacks were evaluated to assess impact on fit. Additionally, one participant applied each fit hack multiple times to assess how reliable hacks were when reapplied. The research of this paper is very important and useful. It can be used to effectively improve the fit of surgical and KN95 masks. The detailed comments are presented below:

1. In Fit hacks: All hacks were tested with two exceptions. One participant was unable to fit the KN95 mask with ear bands tied, as the adjustment caused the wire of the mask to rub against a sore. Another participant was unable to fit pantyhose Brand A over his head. In these two cases, the hack was not tested. Can you provide more details?

In one case the participant had a large and painful acne cyst at the time of testing. Tying the ear bands of the KN95 both adjusted the line of the mask and caused the mask to press tighter so that the edge of the mask was pressing against the cyst. The participant reported the hack was painful and it was thus discontinued.

For the pantyhose test, an approximately 10” section of pantyhose was taken from the thigh of the pantyhose. The pantyhose cross section is an elastic circle. One participant could not fit the pantyhose over their head as the material did not have enough ‘give’ or ‘stretch’.

Some extra detail has been added to the manuscript, however we don’t think that there is much value in providing copious detail.

2. To assess the reliability of these hacks, a further experiment was conducted on one participant. This participant applied each hack five separate times to determine whether the changes in fit from the hack were reproducible when the mask is doffed and donned. The authors say five separate times to determine whether the changes. What's the reference?

This has been edited. We hope this sentence is clearer now.

3. The high degree of variation between applications of the pantyhose hack may indicate that the hack has the potential to tightly seal the mask to the face, but that this potential is not always achieved during every application. Furthermore, fit factor is not scaled linearly with filtration efficiency, and, within higher fit factor values, small changes in filtration efficiency can be represented as larger changes in fit factor. It is not sufficient to present only the results.

Each time the pantyhose was applied; it would compress the mask to the face. However, getting pantyhose over the head was difficult. Sometimes the process of putting it on would disturb the mask’s placement, so that in some cases the mask could become tilted or moved off center. Another thing which influenced fit was hair style. Just putting a ponytail higher or lower could change how tight the pantyhose pressed the mask to the face. We have added this to the manuscript.

4. The conclusion is too simple, a good conclusion can be a restatement of the central idea, a summary of the research results or main points, some revelatory interpretation or consideration, and a prediction based on the research results? It is recommended to rewrite the conclusion.

The conclusion has been rewritten.

Reviewer #2: The manuscript by O’Kelly and colleagues entitled “Face Mask Fit Hacks: Improving the Fit of KN95 Masks and Surgical Masks with Fit Alteration Techniques” aims to assess whether the common “fit hacks” improve the surgical and KN95 mask performance. The notion of improving facemask performance is still an important interventional strategy to reduce SARS-COV-2 transmission as parts of the world are still experiencing serious outbreaks and vaccination campaign still requires time to inoculate naïve populations. I believe this manuscript proves important information of improving mask efficiency and should be published. However, major revision is necessary to answer some questions.

1) Please note there is a difference between a surgical mask (tie) and a procedural mask (earloop). I believe the ones tested in this manuscript are procedural masks.

My understanding is that the term surgical mask is an overarching definition which includes masks which can be secured by tie or earloop. We have now specified that the masks had earloops and added the term ‘procedural masks’.

2) Please indicate the models and manufacturers of the masks tested.

This has been added.

3) The abstract could be more concise, especially the “Results and Discussion” section. I also recommend not using vague words, such as “most” and “many”, repeatedly. How many fit hacks are effective exactly?

The abstract has been edited to make it more specific.

4) Line 65. The claim is not accurate because there are other reasons driving the necessity to wear a “surgical” mask or a KN95 in the public. For example, one would prefer a “surgical” mask than N95 because of comfort? In addition, please explain why not testing cloth masks. They are also widely available to and used by the public.

Each cloth mask has its own shape and there is considerable heterogeneity in the style of these masks. It would be hard to create generalizable findings. In a separate study, we have assessed the performance of home-made fabric face masks.

5) Line 67. Besides fit, material type is also important in filtration efficiency.

This has been noted.

6) The authors have talked a lot about the “fit” of mask and explained what the fit is at line 75. I would suggest expanding the explanation and mentioning it earlier so that it will help readers to understand.

This discussion has been expanded.

7) Line 94-97. I think emphasizing the advantage of quantitative over qualitative fit testing here is not necessary. If the authors really think these are necessary, I believe they should belong to the discussion section.

We do think that highlighting the importance of quantitative fit testing in the methodology over qualitative testing is important as it clarifies to readers that we are not using qualitative methods that they might have encountered when fit testing masks in hospitals. Healthcare facilities often use qualitative fit testing, which relies on users saying whether or not they can taste a test solution which is sprayed around the mask. Since we expect that the article will be of interest to healthcare workers, we wish to clarify that this method (which they may be familiar with) was not the method used.

8) Line 100. What exactly is the standard formula? I think providing the formula or equation will really help readers understand how the quantitative method works, including how the fit factor scores were assigned.

The formula has been added.

9) Line 106: does a Portacount generate real time data output, or just a summary of the whole test results? Is it only giving a “pass” or “fail” result?

Real time data is available and then a summary is provided with the quantitative fit factor. The pass/fail result is based on the quantitative result.

10) Line 110: have the researchers monitored the relative humidity and temperature of the testing environment? I think reporting these data are important since the relative humidity seems to affect the particle count significantly.

Temps were between 65* and 75* F and humidity was between 65% and 70%. We have not included these data in the manuscript as they were not precisely measured at the time of the study. However, article density is taken into consideration by the machine formula. Fit factor is calculated by comparing ambient particles and particles in the mask – it is percentage. 

11) Line 115: The test protocol involves breathing, talking, bending over, and turning the head. Which exact protocol in 29 CFR § 1910.134 was used? Table A-1, A-2, or A-3?

Table A-2

12) The data based on fit factor score is hard to understand without knowing how it was calculated and without a good comparing taget. For example, it will help if a N95 with good fit is tested (assuming >=95% particles were blocked from entry into the mask) and assigned a score.

Please note that this information is present in the manuscript: “When the fit of N95 or FFP3 masks are assessed, a score of at least 100 is required for the mask to be considered to have adequate fit”

13) Another common hack that could improve the mask performance is using a hook to tie the two earloops behind the head. Have you considered and tested this?

We are aware of this hack and use it ourselves in the research group… however, we have unfortunately not yet tested this. Our understanding is that the main purpose hack is used to improve comfort rather than the fit of the mask.

14) Line 274. I thought this paper was mainly to improve mask performance of the masks available and worn by the public. Healthcare workers seem to have no N95 shortage now, at least in the US.

This paper is mainly to help the public, although there is applicability for healthcare workers. Surgical masks and KN95 masks are commonplace in hospitals around the world, with KN95s being prominently used by Asian healthcare workers.

---

## [Decision Letter · Decision Letter 1]

24 Sep 2021

PONE-D-21-11954R1

Face Mask Fit Hacks: Improving the Fit of KN95 Masks and Surgical Masks with Fit Alteration Techniques

PLOS ONE

Dear Dr. O'Kelly,

Thank you for submitting your manuscript to PLOS ONE. After careful consideration, we feel that it has merit but does not fully meet PLOS ONE’s publication criteria as it currently stands. Therefore, we invite you to submit a revised version of the manuscript that addresses the points raised during the review process.

ACADEMIC EDITOR: 

Further revision is necessary.Please address the comments from one reviewer.

We look forward to receiving your revised manuscript.

Kind regards,

Jianguo Wang, PhD

Academic Editor

PLOS ONE

Reviewers' comments:

Reviewer's Responses to Questions

**Comments to the Author**

1. If the authors have adequately addressed your comments raised in a previous round of review and you feel that this manuscript is now acceptable for publication, you may indicate that here to bypass the “Comments to the Author” section, enter your conflict of interest statement in the “Confidential to Editor” section, and submit your "Accept" recommendation.

Reviewer #1: All comments have been addressed

Reviewer #2: (No Response)

2. Is the manuscript technically sound, and do the data support the conclusions?

Reviewer #1: Yes

Reviewer #2: Partly

3. Has the statistical analysis been performed appropriately and rigorously? 

Reviewer #1: Yes

Reviewer #2: Yes

4. Have the authors made all data underlying the findings in their manuscript fully available?

Reviewer #1: Yes

Reviewer #2: Yes

5. Is the manuscript presented in an intelligible fashion and written in standard English?

Reviewer #1: Yes

Reviewer #2: Yes

6. Review Comments to the Author

Reviewer #1: The revisions are satisfactory to this reviewer. The manuscript is recommended for publication in its current form.

Reviewer #2: Please see the attachment. Please see the attachment. Please see the attachment. Please see the attachment.

7. PLOS authors have the option to publish the peer review history of their article (what does this mean?). If published, this will include your full peer review and any attached files.

Reviewer #1: No

Reviewer #2: No

---

## [Author Response · Author response to Decision Letter 1]

20 Oct 2021

We would like to once again thank the reviewers for their thoughtful comments. We have revised the work in response to these, and a letter detailing our changes in response to each comment can be found in this submission.

---

## [Decision Letter · Decision Letter 2]

6 Jan 2022

Face Mask Fit Hacks: Improving the Fit of KN95 Masks and Surgical Masks with Fit Alteration Techniques

PONE-D-21-11954R2

Dear Dr. O'Kelly,

We’re pleased to inform you that your manuscript has been judged scientifically suitable for publication and will be formally accepted for publication once it meets all outstanding technical requirements.

Kind regards,

Jianguo Wang, PhD

Academic Editor

PLOS ONE

Additional Editor Comments (optional):

Reviewers' comments:

Reviewer's Responses to Questions

**Comments to the Author**

1. If the authors have adequately addressed your comments raised in a previous round of review and you feel that this manuscript is now acceptable for publication, you may indicate that here to bypass the “Comments to the Author” section, enter your conflict of interest statement in the “Confidential to Editor” section, and submit your "Accept" recommendation.

Reviewer #1: All comments have been addressed

Reviewer #2: All comments have been addressed

2. Is the manuscript technically sound, and do the data support the conclusions?

Reviewer #1: Yes

Reviewer #2: Yes

3. Has the statistical analysis been performed appropriately and rigorously? 

Reviewer #1: Yes

Reviewer #2: Yes

4. Have the authors made all data underlying the findings in their manuscript fully available?

Reviewer #1: Yes

Reviewer #2: Yes

5. Is the manuscript presented in an intelligible fashion and written in standard English?

Reviewer #1: Yes

Reviewer #2: Yes

6. Review Comments to the Author

Reviewer #1: The revisions are satisfactory to this reviewer. The manuscript is recommended for publication in its current form.

Reviewer #2: The authors have addressed all my comments. This is a great study that will offer impactful information to improve mask performance for the public during the pandemic.

7. PLOS authors have the option to publish the peer review history of their article (what does this mean?). If published, this will include your full peer review and any attached files.

Reviewer #1: No

Reviewer #2: No

---

## [Editor Report · Acceptance letter]

19 Jan 2022

PONE-D-21-11954R2 

Face Mask Fit Hacks: Improving the Fit of KN95 Masks and Surgical Masks with Fit Alteration Techniques 

Dear Dr. O'Kelly:

I'm pleased to inform you that your manuscript has been deemed suitable for publication in PLOS ONE. Congratulations! Your manuscript is now with our production department. 

Kind regards, 

on behalf of

Dr. Jianguo Wang 

Academic Editor

PLOS ONE